# Can Bioelectrical Impedance Analysis and BMI Be a Prognostic Tool in Head and Neck Cancer Patients? A Review of the Evidence

**DOI:** 10.3390/cancers12030557

**Published:** 2020-02-28

**Authors:** Maria Mantzorou, Maria Tolia, Antigoni Poultsidi, Eleni Pavlidou, Sousana K. Papadopoulou, Dimitrios Papandreou, Constantinos Giaginis

**Affiliations:** 1Department of Food Science and Nutrition, School of Environment, University of the Aegean, Myrina, 814 00 Lemnos, Greece; mantzorou.m@aegean.gr (M.M.); elenpav@aegean.gr (E.P.); cgiaginis@aegean.gr (C.G.); 2Radiation Oncology Department, University of Thessaly, School of Health Sciences, Faculty of Medicine, 413 34 Larissa, Greece; mariatolia@uth.gr; 3Department of Surgery, Medical School, University of Thessaly, 413 34 Larissa, Greece; poultsia@yahoo.gr; 4Department of Nutritional Sciences and Dietetics, Hellenic International University, 570 01 Thessaloniki, Greece; souzpapa@gmail.com; 5Department of Health Sciences, CNHS, Zayed University, Abu Dhabi 144534, UAE

**Keywords:** head and neck cancer, bioelectrical impedance analysis, body mass index, weight loss, prognostic factor

## Abstract

*Background:* Malnutrition can significantly affect disease progression and patient survival. The efficiency of weight loss and bioimpedance analysis (BIA)-derived measures in the evaluation of malnutrition, and disease progression and prognosis in patients with head and neck cancer (HNC) are an important area of research. *Method:* The PubMed database was thoroughly searched, using relative keywords in order to identify clinical trials that investigated the role of BIA-derived measures and weight loss on the disease progression and prognosis of patients with HNC. Twenty-seven studies met the criteria. More specifically, six studies examined the prognostic role of the tissue electrical properties in HNC patients; five examined the role of the tissue electrical properties on identifying malnutrition; four studies looked at the changes in the tissue electrical properties of HNC patients; and 12 examined the prognostic role of weight loss on survival and/or treatment outcomes. *Results:* Several studies have investigated the role of nutritional status tools on prognosis in HNC patients. Current studies investigating the potential of BIA-derived raw data have shown that phase angle (PA) and capacitance of the cell membrane may be considered prognostic factors of survival. Weight loss may be a prognostic factor for treatment toxicity and survival, despite some conflicting evidence. *Conclusions:* Further studies are recommended to clarify the role of BIA-derived measures on patients’ nutritional status and the impact of PA on clinical outcomes as well as the prognostic role of weight loss.

## 1. Introduction

Malnutrition is a frequent finding in cancer patients, even at the time of diagnosis. Its incidence varies between 31–87%, depending on disease stage, histopathological type, treatment, and individual patient characteristics [1,2]. Malnutrition can significantly affect disease progression and patient survival. Studies have shown that weight loss in cancer is associated with poor prognosis, poorer quality of life, increased treatment-related adverse effects, and reduced tumor response to treatment as well as lower physical activity levels [3].

Weight loss may develop due to either elevated energy requirements, low energy intake, or compromised nutrient absorption. In cancer patients, undernutrition may be attributed to various factors. In head and neck cancer (HNC) patients, weight loss before therapy is ascribed to several disease-related effects [4]. More to the point, inflammation and catabolism, because of tumor, can lead to muscle wasting and body weight loss [5]. On the other hand, tumor gastrointestinal obstruction can compromise both food intake and absorption as dysphagia, pain, and vomiting can be present. During treatment, eating-related side-effects (such as low appetite, early satiety, nausea and/or vomiting, oral and intestinal mucositis with dysphagia, diarrhea, hemorrhoids, anal fissures, and smell and taste changes) may not only affect total energy intake, but also nutrient absorption, deteriorating nutritional status, while patients’ poor mental health state can diminish their food and energy intake [4,6,7]. Weight loss at diagnosis has been associated with shorter failure-free and overall survival, while being identified as an independent prognostic factor [1,8,9]. In addition to this, weight loss during radiotherapy has been associated with more aggressive disease characteristics [10]. Weight loss at the beginning of chemotherapy is associated with reduced response to treatment and increased toxicity [11,12]. Currently, there is no definitive effective treatment for cancer-associated weight loss and cachexia [13,14], despite decades of research.

Body composition reflects the nutritional status. Bioimpedance analysis (BIA) is based on the body’s tissue electrical properties, and is a non-invasive, time- and cost-effective technique to analyze and monitor body composition [15,16,17]. The principles and applications as well as the drawbacks of BIA have been thoroughly explained by Kyle et al. [16,18] as well as by Sergi et al. [19]. 

BIA measures the *resistance* and *reactance* of the human body by recording the voltage drop in the applied current. The *capacitance* of the cell membranes (Cm/Reactance = 1/2 × π × frequency × capacitance) [20] causes the current to lag behind the voltage, which creates a phase shift, quantified geometrically as the phase angle (PA, phase angle = arc − tangent reactance/resistance × 180°/π) [21,22]. 

More to the point, “the membrane capacitance is proportional to the cell surface area and, together with the membrane resistance, determines the membrane time constant which dictates how fast the cell membrane potential responds to the flow of ion channel currents” [23]. Normal, pre-cancerous cells, and cancer cells have different electrical properties [24]. Human oral cancer cells with higher tumorigenic abilities have exhibited higher Cm [25], while oral cancer progression has been associated with higher Cm of cancer cells [24]. In biological systems, the smaller the quantity of the membranes equals greater capacitance [26]. 

Phase Angle (PA) and bioelectrical impedance vector analysis (BIVA—another graphical method for analyzing BIA raw data [27]) are derived by *reactance* and *resistance* [28]. Both PA and BIVA are considered to reflect both nutritional and hydration status, which are also considered as measures of cell membrane function and integrity [28,29,30].

BIA can be used to assess the body composition of patients of all ages, independently of their physical and mental health status, as this measurement is fast and easily obtainable, with patients only having to step on the scale-analyzer and hold the electrodes. BIA results and raw data are obtained almost immediately, with current body composition analyzers displaying PA in their results. Notably, BIA is currently used in various clinical settings, from hospitals to dietetic clinics, hence it is easy to find a clinician or dietitian that has access to body composition analyzers. Additionally, most body composition analyzers are portable (Table 1). On the other hand, the results of BIA are based on empirical regression equations derived from healthy individuals, who follow a protocol before the measurement, while the different regression equations derived from different populations may not aid in the interpretation of the results [16,18]. The importance of adopting different cutoffs for patient populations has been highlighted [31,32]. Monitoring of each patient’s PA over time has also been suggested [33].

Due to the fact that the results of BIA are based on regression equations for healthy individuals, it has been proposed that raw data, derived by BIA, can be useful to other populations as a nutritional screening and assessment tool, and as a prognostic factor of clinical outcomes [29,34]. PA is considered to be a useful prognostic tool across clinical settings [30,35,36,37], in critical condition patients [38], and especially in cancer patients [33]. It has also been identified as a prognostic factor in colorectal [35] and lung cancer patients [39] as well as in advanced-stage cancer patients [40,41]. 

In addition to this, in healthy adults, PA has been significantly predicted from height, weight, muscle mass, and visceral fat [42], and increases with increasing body mass index (BMI) [43]. During radiotherapy, weight loss has been associated with a decrease in PA [44]. HNC is the seventh most common malignancy worldwide [45]. At diagnosis, 3–52% of HNC patients are categorized as malnourished. During treatment, malnutrition is already present in 44–88% of patients [4,46]. Numerous studies have shown the role of different measures of nutritional status on prognosis and survival in HNC patients [47,48,49,50], highlighting the essential role of nutritional assessment as part of HNC management [4,51]. In this aspect, various studies have been conducted to evaluate the role of easy-to-obtain measures of nutritional status in HNC patients. 

In light of the above considerations, this review paper aims to critically summarize and discuss the currently available clinical data on the efficiency of easily obtainable nutritional status assessment tools such as weight loss and BIA measures in the evaluation of malnutrition in HNC patients, highlighting their role to affect disease progression and prognosis.

The PubMed database was thoroughly searched using relative keywords (weight loss, BIA, Bioimpedance Analysis, head and neck cancer, weight loss, BMI), in order to identify clinical studies that explore the role of BIA-derived raw data and weight loss on disease prognosis and progress as well as highlighting the role of BIA-derived raw data on assessing and predicting malnutrition. Inclusion criteria were:Studies in humans with HNC, where BIA was used to identify malnutrition and/or disease progression and patient prognosis;Studies that investigated the prognostic role of weight loss; andWritten in English language.

Twenty-seven studies met the criteria. More specifically, six studies examined the prognostic role of tissue electrical properties in HNC patients; five examined the role of the tissue electrical properties on identifying malnutrition; four studies looked at the changes in the tissue electrical properties of HNC patients; and 12 examined the prognostic role of weight loss on survival and/or treatment outcomes.

## 2. Bioimpedance Analysis 

### 2.1. The Prognostic Role of Tissue Electrical Properties in Head and Neck Cancer Patients

Currently, there are six studies examining the prognostic role of the tissue electrical properties in HNC patients (Table 2). In general, healthy individuals have a PA between 5° to 7° [28], while a study by Norman et al. identified the fifth percentile of standardized PA as a cutoff point to predict malnutrition, functionality, quality of life, and mortality in cancer patients [52]. 

Wladysiuk et al. [53] showed that PA was a prognostic factor in HNC. Specifically, 75 HNC patients with disease stage IIIB and IV underwent BIA at 50 kHz in order for PA to be measured. The risk of shorter overall survival was significantly higher in patients presenting PA less than 4.7°, compared to those with higher PA [53]. Accordingly, Buntzel et al. [54] carried out a retrospective study to investigate the prognostic role of BIA data on disease outcome. Clinical data were collected from 66 advanced HNC patients who were receiving nutritional therapy and radiotherapy. Twenty-seven patients survived and 39 died between entry measurement and the last measurement. BIA was performed every four weeks during patient visits, after finishing baseline treatment. Survivors exhibited a stabilized PA (4.7° to 5.2°), while deceased patients had a significantly lower PA (4.6° to 3.7°) [54]. Similarly, Buntzel et al. [55] in their recent study found that HNC patients with normal PA (>5.0) had longer survival (13.84 months vs. 51.16 months). Additionally, Axelsson et al. in their retrospective study of 128 advanced HNC patients also found that a PA cutoff of 5.95° was prognostic for five-year overall survival, while being an independent prognostic factor for survival [56].

Another recent study, with 61 patients, highlighted the fact that low PA was associated with malnutrition as well as longer hospital stay and complications in HNC patients [57]. 

Considering Cm, Malecka-Massalska et al. [20] performed a study to evaluate the prognostic significance of Cm in HNC patient survival. For this purpose, Cm measurements via BIA at 50 kHz was performed in 75 advanced HNC patients (stages IIIB and IV). Notably, this study found that well-nourished patients had significantly higher median Cm compared to malnourished ones. A receiver operating characteristics (ROC) curve analysis with a cut-off value of 0.743 was obtained. This was characterized by 98% specificity and 37% sensitivity in the detection of malnutrition. Moreover, patients with Cm below the cutoff value were at greater risk of shorter overall survival compared to those presenting higher Cm. Importantly, Cm was identified as a strong, independent prognostic factor for overall patient survival in HNC [20].

### 2.2. The Role of Tissue Electrical Properties on Identifying Malnutrition

Five studies have currently evaluated the role of tissue electrical properties on identifying malnutrition in HNC patients, two of which also investigated the role of PA on predicting malnutrition (Table 3). More to the point, in a prospective cohort study [58], HNC patients were categorized as well-nourished or malnourished using a validated nutritional assessment tool Subjective Global Assessment (SGA) [59,60]. PA was measured in 75 patients with histologically confirmed HNC. This study supported evidence that well-nourished patients had a significantly higher median PA of 5.25°, compared to malnourished ones who had a median PA of 4.73°. A PA cut-off of 4.73° was 80% sensitive and 56.7% specific in detecting malnutrition, diagnosed by SGA [58]. In a recent study by Lundberg et al. [61] conducted in 41 newly diagnosed HNC patients, despite the normal average BMI of 25.2 kg/m^2^, low fat-free mass index was seen in 44% of female and in 28% of male patients. Moreover, PA was lower than the reference values in the vast majority (76%) of patients. With the use of BIVA, 32% of patients were within the normal range, while 37% were found to be malnourished. As a result, it was proposed that BIA at presentation could be a practical method to detect malnutrition, also analyzing body composition and identifying high-risk HNC patients [61].

On the other hand, a recent study by Stegel et al., [62] documented contradictory results. In this study, the nutritional status of 55 HNC patients was measured by Nutritional Risk Screening 2002, anthropometric and laboratory tests as well as BIA before and after chemo-radiotherapy. Cachexia was diagnosed by the international consensus criteria and patients were categorized as well-nourished, malnourished or cachectic. More to the point, it was found that the patients’ nutritional status worsened after treatment. Notably, well-nourished patients had a higher pre-treatment mean PA compared to malnourished and cachectic counterparts. In fact, the risk of malnutrition and cachexia increased by 71% per mean PA decrease by one-unit. However, pre-treatment PA failed to show any predictive value for cachexia during therapy, and it did not distinguish malnourished from cachectic HNC patients. Despite the above, PA was considered as a good marker of nutritional status in this patient population [62].

The utility of PA in detecting malnutrition was also evaluated by Mulasi et al. [63]. Using the Academy of Nutrition and Dietetics and American Society for Parenteral and Enteral Nutrition Consensus malnutrition definition, the authors estimated the prevalence of malnutrition in a sample of HNC patients and compared it to the validated tool patient generated-SGA (PG-SGA). They also investigated the utility of 50-kHz PA, and 200-kHz/5-kHz impedance ratio to diagnose malnutrition. For this purpose, 18 male and one female patient scheduled to undergo chemo-radiotherapy were seen five times during treatment and up to three months post-treatment. Multi-frequency BIA, PG-SGA, physical examination for nutrition-related parameters, anthropometric indices, dietary intake, and handgrip strength data were recorded. Using the consensus malnutrition definition, 67% of the patients were found to be malnourished before treatment. Malnourished patients, as diagnosed by the consensus criteria (but not the PG-SGA), had a lower mean PA and higher impedance ratio than well-nourished ones. Both PA and impedance ratio were correlated with higher PG-SGA score and handgrip strength, but their clinical utility on nutritional status and muscle loss was unclear [63]. Di Renzo et al., however, found that PA was useful not only as a prognostic factor for survival, but also as a means of evaluating progress after nutritional intervention in malnourished patients with stage III HNC [64].

### 2.3. Bioimpedance Analysis Data on Head and Neck Cancer 

Four studies have evaluated the changes in the tissue electrical properties of HNC patients (Table 4). From this aspect, De Luis et al. [65], in their case-control study, were the first to investigate the differences of PA and other impedance parameters in HNC patients. Sixty-seven ambulatory post-surgical male patients with oral and/or laryngeal cancer, without a recent weight loss (defined as <5% WL the past three months) were enrolled, along with a matched-control group of 70 healthy males. Notably, anthropometric evaluation showed that both mean fat mass and fat-free mass were lower in cancer patients compared to healthy individuals. Moreover, both reactance and PA were lower in cancer patients compared to healthy individuals. Despite the normal weight and BMI between the two groups, considerable altered tissue electrical properties were observed. It is worth noting though that in this study, HNC patients had a higher PA compared to those of other studies, which may be ascribed to the fact that measurements were taken occasionally after surgery, when patients’ nutritional status could be improved [65].

Following the above study, Malecka-Massalska et al. investigated the altered electrical tissue properties in HNC via BIA. In fact, the authors [66] performed a cross-sectional study to explore the tissue electrical properties in HNC patients. In this study, 31 patients and 31 age- and sex-matched healthy controls were enrolled. PA was statistically significantly lower, and the resistance was statistically significantly higher in HNC patients compared to the control group [66]. In another substantial study, Malecka-Massalska et al. [67] applied BIA to assess the changes to the tissue electrical properties in HNC patients. Notably, this study aimed to examine the soft tissues’ hydration and mass through the pattern analysis of vector plots as height, normalized resistance, and reactance measurements via BIVA. It should be noted that patients were untreated and were not under nutritional intervention. Twenty-eight white, male HNC patients, and 28 healthy individuals matched by sex, age, and BMI were included in the study. Mean vectors of patients, compared to healthy counterparts, had an increased normalized resistance with a reduced reactance, indicating dehydration with loss of dielectric mass (“cell membranes and tissue interfaces”) of soft tissues. Hence, this study supported evidence that BIVA may be a useful tool to prevent post-operational complications [67]. Moreover, Malecka-Massalska et al. [68], in a later study, used the same protocol as above, using BIA measures to extract PA and BIVA in 67 white, male, HNC patients (22–87 years old) and 67 matched healthy individuals were enrolled. Again, patients were untreated and were not under nutritional intervention. Mean vectors of cancer patients were characterized by increased normalized resistance and decreased reactance compared to healthy counterparts. The above findings were in accordance with those of the former study. BIVA was also considered to be an objective measure that can help improve decision-making in a clinical setting, and assess prognosis [68].

## 3. Weight Loss

### The Prognostic Role of Weight Loss on Survival and/or Treatment Outcomes

There are currently 12 studies evaluating the prognostic significance of weight loss on survival and treatment outcomes of HNC patients (Table 5). HNC is a heterogenous group of cancers that affect food intake in different ways, which can lead to different outcomes.

In a retrospective study, Buntzel et al. [54] investigated the outcome of 110 HNC patients in relation to initial weight loss at diagnosis as well as weight loss at the end of radiotherapy. An initial weight loss of ³10 kg was related to higher mortality, while a total weight loss of ³15 kg at the end of baseline therapy was recorded [54]. In another study, Du et al. [69] evaluated the impact of the Prognostic Nutritional Index and weight loss on metastasis and long-term mortality in nasopharyngeal carcinoma (NPC) patients by reviewing data of 694 newly diagnosed NPC patients. Greater weight loss (≥10% initial body weight) was an independent prognostic factor for poor overall and distant metastasis-free patient survival [69]. Shen et al. [70] investigated the prognostic role of weight loss and whether BMI could mediate this effect. From this aspect, 2433 NPC patients under definitive radiotherapy were examined. Weight loss during treatment was classified into two categories: high (>5% weight loss) and low (<5% weight loss). Among the underweight and normal weight patients, high weight loss was independently associated with poorer overall and disease-specific survival compared to those with low weight loss. However, in overweight/obese patients, no significant associations between high weight loss and overall or disease-specific survival were recorded. Hence, except for overweight and obese patients, high weight loss during radiotherapy was an independent prognostic factor of shorter survival, especially in underweight patients [70]. Another study performed by Ghadjar et al. [71] evaluated the prognostic impact of weight loss before and during chemo-radiation in 224 locally advanced HNC patients. Weight was measured six months before treatment initiation, at the beginning, and at the end of the treatment. After a median follow-up of 9.5 years, pre-treatment weight loss was independently associated with greater treatment failure, shorter loco-regional recurrence-free survival, and shorter distant metastasis-free survival as well as cancer specific and overall survival. However, weight loss during treatment was not associated with patient survival [71]. 

Ehrsson et al. [72] explored the role of treatment, tumor site and stage, BMI, gender, age and civil status in predicting weight loss, with the aim to explore potential associations between weight loss on post-operative infections and mortality. One-hundred and fifty-seven HNC patients were enrolled and followed-up for two years after radiotherapy. Demographic, disease-specific, and nutrition data were collected from patient records. Tumor stage was the only independent factor of the greatest weight loss, whereas weight loss was not associated with post-operative infection risk and/or mortality [72]. 

In a retrospective study, Grossberg et al. [73] further evaluated the role of baseline and post-treatment body composition on disease outcome with the aim to determine if lean body mass before and after radiotherapy can predict survival in HNC patients. One-hundred and ninety HNC patients, subjected to whole-body positron emission tomography-computed tomography (CT) or abdominal CT scans before and after radiotherapy, were examined. Skeletal muscle depletion was detected in 67 patients before radiotherapy and a further 58 patients after radiotherapy. Diminished skeletal muscle mass, measured by CT imaging and BMI, were identified as predictors of oncologic outcomes in this patient population, but weight loss after radiotherapy initiation did not predict the patients’ skeletal muscle loss or survival [73].

Additionally, a retrospective study by Ottosson et al. [74] evaluated the role of weight loss and BMI in relation to five-year overall survival in oropharyngeal cancer patients. In this study, nutritional data were based on percentage weight loss from the start of radiotherapy and up to five months after radiotherapy for 232 patients as well as from data concerning patients’ BMI at the start of radiotherapy for another 203 patients. Notably, BMI at the start of radiotherapy was a prognostic factor for five-year overall survival in this study population, whereas weight loss was not significant [74]. In addition, De Cassia Braga Ribiero et al. [75] assessed the role of various clinical factors as potential prognostic factors of peri-operative complications and mortality in oral and oropharyngeal carcinoma patients. In fact, 530 patients submitted to surgical treatment were enrolled in this study. Weight loss was found to exert a significant effect on patients’ prognosis in univariate analysis, but not in multivariate analysis [75]. 

Designing a prospective study, Karnell et al. [76] evaluated the impact of pre-treatment BMI and three-month weight changes on the survival of 578 HNC patients. This study documented that higher BMI was associated with better survival, while patients with stable weight had the highest five-year survival rate (72.6%). Moreover, patients with weight loss ≥5% had worse survival (45.8%) compared to those with <5% weight loss (65.8%). Although BMI independently predicted patient survival, weight change was not identified as an independent prognostic factor [76]. In addition, Lin et al. [77] retrospectively evaluated the prognostic role of BMI and weight loss on therapeutic outcome of NPC patients who only received intensity-modulated radiation therapy as treatment. In this study, 34% out of 238 patients had a pre-treatment BMI of ≥23 kg/m^2^ and 63% of them had significant weight loss (≥5%). Notably, this study documented that patients with BMI ≥ 23 kg/m^2^ did not have a better three-year overall, disease-specific, locoregional free, or distant metastatic free survival. Moreover, patients with significant weight loss did not have worse three-year clinical endpoints, even after adjusting for BMI and after the sensitivity test. Hence, this study did not find any significant relationship between BMI and weight loss on the survival of this patient population [77].

Concerning the prognostic role of weight loss on treatment toxicity, Meyer et al. [78] investigated potential prognostic factors for radiotherapy-related acute and late toxicities in 540 radiotherapy treated HNC patients. Treatment adverse effects were assessed using the Radiation Therapy Oncology Group Acute Radiation Morbidity Criteria during, and one-month after radiotherapy as well as the Radiation Therapy Oncology Group/European Organization for Research and Treatment of Cancer Late Radiation Morbidity Scoring Scheme at six and 12 months after radiotherapy. Interestingly, weight loss during radiotherapy was identified as an independent prognostic factor for severe late toxicity [78]. Analogous findings were also recorded by Ghadjar et al. [79]. In fact, this study aimed to identify predictive factors for severe late radiotherapy-related toxicity after hyper-fractionated radiotherapy treatment with or without concomitant cisplatin in locally advanced HNC [79]. Patient data were retrospectively investigated from a randomized phase III trial (SAKK 10/94). Data from 213 randomized patients were analyzed, of whom 39% experienced severe late radiotherapy-related toxicity. Weight loss ratio, along with advanced N-classification, unresectable tumor, and severe acute dysphagia were identified as independent prognostic factors for severe late toxicity due to radiotherapy [79].

## 4. Conclusions

As far as BIA is concerned, the majority of currently available studies have shown a good potential on its clinical use in HNC, in accordance with studies in other cancer patient populations [32,35,39,40,41,80]. Until now, it should be noted that few studies have investigated the prognostic potential of the raw data derived by BIA in this patient population, even though they can be easily obtained. The aforementioned studies have shown that both PA and capacitance of the Cm may be considered to be prognostic factors of patient survival in HNC. On the other hand, there are some studies that have documented contradictory results concerning the potential of BIA and BIVA on predicting patients’ malnutrition. Even though three of the five studies reported encouraging results for the prediction of malnutrition, the exact role of PA on detecting malnutrition was deemed unclear in one study, and there was no predictive value in another. Moreover, in another study, PA could not differentiate malnourished from cachectic patients, while pre-treatment PA was not a predictive factor for cachexia during chemo-radiation treatment. In addition to this, a review by Rinaldi et al. showed that currently, in cancer patients, the agreement between PA and the SGA tool regarding malnutrition, is weak [81]. Thus, further studies are needed in order to assess the association between malnutrition and altered tissue electrical properties as well as its subsequent prognostic role in HNC patients. Future studies evaluating the efficiency of BIA raw data may use more advanced body composition analyzers than those used in the currently available studies, since these analyzers are more accurate and display PA in their results. However, it is important to adopt specific cut-off points for PA for each patient population.

Considering weight loss, several studies have investigated its prognostic impact on patient survival and post-operative complications. Some studies have supported its prognostic value; however, other studies have shown a lack of prognostic capacity for weight loss in this patient population. It is important to note the heterogeneity of HNC in terms of affecting food intake, and thus outcomes regarding weight loss as well as the important role of cancer stage when analyzing the results. Great weight loss of >10% of initial body weight during treatment was found to be an independent predictive factor for one-year overall, disease-free, and disease-specific patient survival. Some studies have documented that pre-operative weight loss >10% was associated with greater five-year mortality, while pre-treatment weight loss was an independent prognostic factor for greater treatment failure, shorter locoregional recurrence-free survival, and distant metastasis-free survival as well as disease-specific and overall patient survival. On the other hand, other studies have failed to find an association between weight loss (at the beginning or after radiotherapy) and patient post-operative outcomes and survival. Additionally, the two identified studies that concern treatment toxicity showed that weight loss and weight ratio were independent prognostic factors for late severe treatment-related toxicity.

When taking BMI into account, one study found that, except for patients categorized as overweight and obese, high weight loss during radiation treatment was independently associated with poor nasopharyngeal carcinoma patient survival, with the impact being more prominent in underweight patients. In addition, another study in the same patient population did not find such an association as patients with significant weight loss did not have worse three-year clinical endpoints, even after adjusting for the impact of weight loss by BMI category.

In the past decades, tissue electrical properties have been studied in healthy individuals and patient populations. Adjunct BIA and weight loss monitoring are non-invasive, easy-to-use, and promising tools regarding the screening and assessment of the nutritional status of HNC patients, with prognostic value. A number of studies in other cancer patient populations highlight the prognostic role of PA and the utility of BIA [35,39,52,82,83], European Society fpr Parenteral and Enteral Nutrition (ESPEN) guidelines suggest the use of BIA in patients undergoing hematopoietic stem cell transplantation [84].

It is important to note that the majority of enrolled patients in the currently available studies were men, possibly due to the fact that HNC frequently effects more men than women [85]. However, future studies also need to address the role of different prognostic factors on clinical outcomes in women with HNC. More studies are also recommended in order to clarify the role of BIA-derived measures assessing nutritional status in HNC patients. Moreover, it is also important to examine the impact of deteriorating PA and BIVA measurements on patient survival and other crucial clinical outcomes such as post-operative complications and treatment-related toxicity. In addition to this, further studies can also focus on regression equations for cancer patient populations. Moreover, additional high-quality and well-designed studies, which take into account body composition and BMI, should be performed to accurately clarify the prognostic role of weight loss in HNC.

## Figures and Tables

**Table 1 cancers-12-00557-t001:** The advantages of using bioimpedance analysis in cancer patients.

Why Use Bioimpedance Analysis in Cancer Patients?
• Easy to use • Time-effective • Cost-effective • Non-invasive
• Raw data can be easily obtained immediately (Phase Angle, Reactance, and Resistance are displayed in the results of the new models of body composition analyzers)
• Available in various clinical settings (hospitals and dietetic clinics)
• Portable devices available
• New models function as both scales and body composition analyzers
• Feasible for patients of all ages, physical and mental health states

**Table 2 cancers-12-00557-t002:** Studies examining the prognostic role of tissue electrical properties in head and neck cancer patients.

Study Type	Patient Population (n = Number of Participants)	Outcomes	Reference
Prospective study	Stage IIIB and IV patients with HNC (n = 75)	Significantly ↑ risk of shorter OS in patients with PA < 4.7°, compared to those with higher PA.	[53]
Retrospective study	Advanced HNC patients under nutritional therapy and radiotherapy (RT) (n = 66)	Survivors had a stabilized PA (4.7° to 5.2°) and deceased patients had a significantly lower PA (4.6° to 3.7°)	[54]
Retrospective study	HNC patients (n = 42)	Patients with normal PA > 5.0 had a significantly better survival (13.84 vs. 51.16 months)	[55]
Retrospective study	Patients with advanced HNC (n = 128)	PA at diagnosis a significant factor for survival Cutoff point for 5-year survival is 5.95°	[56]
Retrospective study	HNC patients (n = 61)	HNC patients have a low PA at diagnosis. Low PA is associated with a long hospital stay and complications.	[57]
Prospective study	Stage IIIB and IV HNC patients (n = 75)	↑risk of shorter survival with Cm below 0.743 compared to patients with higher Cm. Cm was a strong, independent prognostic factor of OS in HNC	[20]

HNC: Head and Neck Cancer, PA: Phase Angle, Cm: Capacitance of cell membrane, OS: Overall Survival.

**Table 3 cancers-12-00557-t003:** Studies examining the role of tissue electrical properties on identifying malnutrition.

Study Type	Patient Population (n = Number of Participants)	Outcomes	Reference
Prospective study	Patients with histologically confirmed HNC (n = 75)	Well-nourished patients (according to SGA) had a significantly ↑ median PA (5.25°) compared to malnourished patients (4.73°)	[58]
Prospective study	Newly diagnosed HNC patients (n = 41)	BIA at presentation was a practical method to detect malnutrition, analyze body composition and identify high-risk HNC patients	[61]
Prospective study	HNC patients (n = 55)	Well-nourished patients had a ↑ pre-treatment PA ↑ risk of malnutrition/cachexia with ↓ PA PA did not distinguish malnourished from cachectic patients. PA a good marker of nutritional status in this patient population	[52]
Prospective study	HNC patients undergoing chemoradiotherapy and up to 3 months after treatment completion (n = 19)	PA and impedance ratio were correlated with higher PG-SGA score and handgrip strength. Unclear if they could be used as surrogate markers of nutrition status or muscle loss.	[63]
Prospective study	Malnourished HNC patients, at stage III (n = 50)	PA a predictor of cancer survival PA useful in the surveillance of nutritional status improvement and biochemical indices.	[64]

HNC: Head and Neck Cancer, BIA: Bioimpedance Analysis, PA: Phase Angle, PG-SGA: patient generated-subjective global assessment, SGA:subjective global assessment.

**Table 4 cancers-12-00557-t004:** Studies examining changes in the tissue electrical properties of head and neck cancer patients.

Study Type	Patient Population (n = Number of Participants)	Outcomes	Reference
Case-control study	Weight stable, ambulatory post-surgical male patients with oral and/or laryngeal cancer (n = 67) with a matched control group of (n = 70)	Reactance and PA were lower in cancer patients than in controls Altered tissue electrical properties were observed in the patient group	[65]
Case-control study	HNC patients (n = 31) and healthy age and sex matched controls (n = 31)	↓ PA in HNC patients ↑Resistance in HNC patients	[66]
Case-control study	SCCHN patients (n = 28) and matched control group (n = 28)	↑ Resistance and ↓Reactance in patient group Results indicate dehydration with loss of with loss of dielectric mass of soft tissue BIVA can be used to prevent post-operational complications	[67]
Case-control study	HNC patients (n = 67) and matched control (n = 67)	↑ Resistance and ↓ Reactance in patient group Results indicate dehydration with loss of with loss of dielectric mass of soft tissue BIVA offers objective measures to improve clinical decision-making and predicting outcomes	[68]

HNC: Head and Neck Cancer, PA: Phase Angle, SCCHN: squamous cell carcinoma of the head and neck.

**Table 5 cancers-12-00557-t005:** Studies examining the prognostic role of weight loss on survival and/or treatment outcomes.

Study Type	Patient Population (n = Number of Participants)	Outcomes	Reference
Retrospective study	SCCHN patients (n = 110)	↑ mortality after: initial weight loss of >10 kg, total weight loss of >15 kg at the end of baseline therapy	[54]
Retrospective study	newly diagnosed NPC patients (n = 694)	WL of >10% of initial body weight was an independent predictor of poor OS, and Distant Metastasis-Free Survival	[69]
Prospective study	NPC patients receiving radical radiotherapy (n = 2433)	Except for overweight/obese patients, high weight loss during radiation was independently associated with poor survival	[70]
Prospective randomized study	Patients with locally advanced SCCHN (n = 224)	Weight loss before treatment was independently associated with treatment failure, ↓Locoregional Recurrence-Free Survival and ↓ Distant Metastasis-Free Survival, ↓ cancer specific survival and ↓ OS	[71]
Retrospective single-institution cohort study	HNC patients (n = 157)	Weight loss was not significantly related to risk for post-operative infection and mortality	[72]
Retrospective study	SCCHN patients undergoing curative RT (n = 190)	Weight loss after radiotherapy initiation cannot predict skeletal muscle loss or survival	[73]
Retrospective, randomized, multicenter ARTSCAN trial	Oropharyngeal cancer patients (n = 232)	Weight loss was not a prognostic factor for 5-year OS	[74]
Prospective study	Oral and oropharyngeal cancer patients (n = 530)	Weight loss was found to have an impact on prognosis, in univariate analysis, but not in multivariate analysis	[75]
Prospective study	HNC patients (n = 578)	Patients with stable weight had the highest 5-year survival rate Patients who gained ≥5% had worse survival than those who lost ≥5% Weight change was not an independent predictor of survival	[76]
Retrospective study	NPC patients who received intensity-modulated radiation therapy (n = 238)	No significant relationship between BMI and percent weight loss on survival	[77]
Prospective study	Patients treated with RT for localized HNC (n = 540)	Weight loss during RT was an independent predictor for severe late toxicity	[78]
Randomized phase III trial: SAKK 10/94	Patients with locally advanced HNC (n = 213)	Weight loss ratio, was independent prognostic factor for severe late RT-related toxicity	[79]

HNC: Head and Neck Cancer, NPC: Nasopharyngeal Carcinoma, SCCHN: Squamous Cell Carcinoma of the Head and Neck, BMI: Body Mass Index, RT: radiotherapy.

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
