# Peer review of "Can Bioelectrical Impedance Analysis and BMI Be a Prognostic Tool in Head and Neck Cancer Patients? A Review of the Evidence"

_cancers, 2020, doi:10.3390/cancers12030557_

Round 1

Reviewer 1 Report

A well written review.

Some comments:

line 21 ond others: disease progression (not progress)

line 39-41: I wold add more references about the association of weight loss and prognosis

line 49-50: the verb "to affect" is repeated in the same period and the role of mental healt state should be clarified

line 85: inclusion criteria are not well defined, they cannot simply be studies in humans with KNC and written in english

line 133: SGA, it would be useful a reference, where it comes from?

line 202: 13 studies, but table 5 report only 12 studies and, in the table, the study bi Ottosson is retrospective, not prospective

line 213: I would suggest definitive radiotherapy (not radical)

line 299-300: "decreased treatment failure locoregional recurrence free survival" is a long phrase I can hardly understand. Do you mean simply "decrease recurrence-free survival"?

Author Response

All comments have been addressed and described in the attached file.

Reviewer 2 Report

The review attempted to summarize findings in HNC clinical trials which can be directly or indirectly “indicating the potential role of BIA-derived measures and weight loss on HNC disease progression and prognosis. It summarizes findings from some HNC correlative studies. The review was written with lots of jargons without definitions.  The serious lack of clear definitions for many technical terms has made this review hard to follow.

Many grammatical errors throughout the text, please kindly fix them. e.g. the terms “disease progress” and “patients’ survival” should be changed to “disease progression” and “patient survival” Authors need to define what exactly BIA is in the Introduction (around line 53-59). There is no clear definition of BIA, while the text suddenly jumped to the advantages of BIA. Again, the text (line 61-62) suddenly jumped to BIA regression equations…… What are these exactly?? The introduction is not well-structured. If the entire review is on BIA, the authors should first defined BIA in details, then application, advantages and disadvantages, etc. I believe it is better to restructure the introduction for clarity. Otherwise, it is confusing. Line 71: Head and neck cancer is the 6th most common cancer Various terms are being used in the text for head and neck cancer: HNSCC and HNC . please unify the terms Line 94: healthy individuals have a PA between 5° to 7°. What does this sentence mean?? If the authors use jargon within the field, these jargons have to be fully explained to general readers first in the text. Lines 98 , 121: “in this aspect”?? what aspect?? May not be appropriate to use. Line 119-120: unclear what it means. Please explain more clearly. Please explain what contributes to Cm? Need to define what is SGA. Weight loss is a “compound consequences” resulting from many factors. How can BIA changes be specifically linked to the change in a specific factor in HNC patients? Any findings addressing this issue? How is BIA related to BMI?? After summarizing all these findings, what do the authors want to emphasize? What else should be improved for BIA-related studies if it would ever be adopted to predict disease progression or other clinical outcomes, etc.? Are there any success in other cancer types in adopting BIA-related parameters/measures for patient outcomes?

Author Response

All comments have beed addressed and described in the attached file.

Round 2

Reviewer 2 Report

All comments have been addressed. Thank you.

Author Response

The manuscript underwent a language editing by a native English speaker. All changes are noted as a track change in the revised paper.